# Functional exploration of heterotrimeric kinesin-II in IFT and ciliary length control in *Chlamydomonas*

**Shufen Li[1,2], Kirsty Y Wan[3], Wei Chen[1], Hui Tao[1], Xin Liang[1]\*, Junmin Pan[1,2]\***

[1]MOE Key Laboratory of Protein Sciences, Tsinghua-Peking Center for Life Sciences, School of Life Sciences, Tsinghua University, Beijing, China; [2]Laboratory for Marine Biology and Biotechnology, Qingdao National Laboratory for Marine Science and Technology, Qingdao, China; [3]Living Systems Institute, University of Exeter, Exeter, United Kingdom

**Abstract** Heterodimeric motor organization of kinesin-II is essential for its function in anterograde IFT in ciliogenesis. However, the underlying mechanism is not well understood. In addition, the anterograde IFT velocity varies significantly in different organisms, but how this velocity affects ciliary length is not clear. We show that in *Chlamydomonas* motors are only stable as heterodimers in vivo, which is likely the key factor for the requirement of a heterodimer for IFT. Second, chimeric CrKinesin-II with human kinesin-II motor domains functioned in vitro and in vivo, leading to a ~ 2.8 fold reduced anterograde IFT velocity and a similar fold reduction in IFT injection rate that supposedly correlates with ciliary assembly activity. However, the ciliary length was only mildly reduced (~15%). Modeling analysis suggests a nonlinear scaling relationship between IFT velocity and ciliary length that can be accounted for by limitation of the motors and/or its ciliary cargoes, e.g. tubulin.

**\*For correspondence:**
xinliang@tsinghua.edu.cn (XL);
panjunmin@tsinghua.edu.cn (JP)

## Introduction

It is well established that cilia are conserved cellular organelles that play pivotal roles in signaling and cell motility, and defects in cilia are linked with numerous human diseases and developmental disorders (*Anvarian et al., 2019*; *Bangs and Anderson, 2017*; *Reiter and Leroux, 2017*). The assembly and maintenance of cilia require intraflagellar transport (IFT), a bidirectional movement of protein complexes (IFT complexes) between ciliary membrane and the axoneme (*Kozminski et al., 1993*). The anterograde transport (from ciliary base to tip) is driven by kinesin-2 whereas retrograde transport (from ciliary tip to base) is powered by cytoplasmic dynein 2/1b (*Rosenbaum and Witman, 2002*; *Scholey, 2003*). IFT complexes, which consist of IFT-A and IFT-B complexes, serve as cargo adaptors to recruit ciliary proteins (*Lechtreck, 2015*; *Taschner and Lorentzen, 2016*), and are assembled into linear arrays termed IFT particles or IFT trains (*Kozminski et al., 1993*; *Pigino et al., 2009*).

Heterotrimeric kinesin-2 (kinesin-II) is essential for anterograde IFT and ciliogenesis in most ciliated cells while both heterotrimeric and homodimeric kinesin-2 collaboratively drive anterograde IFT in *C. elegans* (*Scholey, 2013*). In contrast to most kinesins with two identical motor subunits, kinesin-II consists of two non-identical motor subunits and one non-motor subunit (KAP) (*Hirokawa et al., 2009*; *Verhey and Hammond, 2009*). The heterotrimeric organization of kinesin-II is required for IFT because mutation in either subunit abolishes or impairs IFT in various organisms (*Engelke et al., 2019*; *Kozminski et al., 1995*; *Liang et al., 2014*; *Lin et al., 2003*; *Miller et al., 2005*; *Mueller et al., 2005*; *Nonaka et al., 1998*; *Snow et al., 2004*). This may result if a homodimer of the motor subunits cannot be properly formed and/or a homodimer cannot associate with KAP or

IFT complexes. For example, in *C. elegans*, homodimer of kinesin-II motor subunits cannot be formed in vitro and motors with two identical motor domains are not functional in vivo (*Brunnbauer et al., 2010*; *Pan et al., 2010*). In mammalian cells, by over-expressing KIF3A or KIF3B it was shown that KIF3A homodimers can form but it cannot associate with IFT complexes whereas KIF3B cannot form homodimers (*Funabashi et al., 2018*). It is intriguing to learn how this heterotrimeric organization requirement is conserved and diversified, especially in the unicellular eukaryote *Chlamydomonas* in which IFT was first discovered (*Kozminski et al., 1993*).

During IFT, the velocity of anterograde IFT driven by kinesin-II varies from organism to organism. Though the assay conditions would slightly affect the measurements, the velocity of anterograde IFT is ~2.2 µm/s in *Chlamydomonas* and Trypanosome (*Bertiaux et al., 2018a*; *Brown et al., 2015*; *Dentler, 2005*; *Engel et al., 2009*; *Liang et al., 2014*; *Wingfield et al., 2017*). In contrast, mammalian cells and worms have a much slower velocity (~0.5 µm/s) (*Broekhuis et al., 2014*; *Engelke et al., 2019*; *Follit et al., 2006*; *Snow et al., 2004*). Notably, *Chlamydomonas* and Trypanosoma have longer cilia whereas mammalian cells tend to have shorter cilia. It is intriguing how motor speed affects IFT, ciliary assembly and, in turn, controls ciliary length.

In this work, we reveal distinct mechanisms for the requirement of the heterodimeric motor organization of *Chlamydomonas* kinesin-II (CrKinesin-II), especially that the stability of the two motor subunits depends on each other in vivo. Furthermore, we generated chimeric CrKinesin-II with motor domains of human kinesin-II (HsKinesin-II) and show that it can perform motility function in vitro and in vivo but with an ~2.8 fold reduction in the velocity of motor and anterograde IFT. The reduced motor velocity results in a similar reduction in the IFT injection rate. IFT injection rate has been shown to correlate with ciliary assembly activity and thus ciliary length (*Engel et al., 2009*; *Marshall et al., 2005*; *Marshall and Rosenbaum, 2001*). However, the effect of changing motor speed on ciliary length has not been directly demonstrated in vivo. Interestingly, our results reveal that the ciliary length of the cells expressing slow chimeric motors is only mildly reduced (~15%). Using a modeling approach to understand the effect of motor speed on ciliary assembly and length, we reveal that limitation of motors is likely the key determinant of ciliary length. As an extension of our modeling analysis, the potential effects of limitation of key ciliary components, for example tubulin, and a length-dependent depolymerization rate on ciliary length are also discussed.

## Results

### The requirement of heterotrimeric organization of CrKinesin-II for IFT

The function of kinesin-II in IFT requires two non-identical motor subunits and a non-motor subunit, kinesin-associated protein (KAP). To understand how this organization is required for the function of CrKinesin-II in IFT, we analyzed whether CrKinesin-II with two identical motor domains can coordinate for motility in vitro and in vivo, and whether each of the motor subunits can interact with KAP independently without the other subunit. For the in vitro motility assay, we generated CrKinesin-II constructs with fluorescent tags and purification tags (*Figure 1A*). To generate CrKinesin-II constructs with two identical motor domains, the motor domain of FLA8 was replaced with that of FLA10 and *vice versa* (*Figure 1B*). The recombinant wild-type CrKinesin-II as well as the chimeric motors FLA10/FLA10'/KAP and FLA8/FLA8'/KAP were expressed respectively in Sf9 cells and purified (*Figure 1—figure supplement 1*).

We used total internal reflection fluorescence (TIRF) microscopy to determine the motility of the purified motors. Compared to wild-type motors (1.62 ± 0.23 µm/s, n = 55), FLA8/FLA8'/KAP (1.60 µm/s ± 0.17, n = 83) moved with a similar velocity while FLA10/FLA10'/KAP (1.88 ± 0.20 µm/s, n = 56) showed a slightly higher velocity. Thus, chimeric motors with two identical motor domains (i.e. FLA10/FLA10 and FLA8/FLA8) of CrKinesin-II can functionally coordinate in vitro. These results were consistent with the reports for kinesin-II from *C. elegans* and mammal (*Brunnbauer et al., 2010*; *Muthukrishnan et al., 2009*; *Pan et al., 2010*). However, kinesin-II with two identical motor domains of KLP20 in *C. elegans* did not function in vivo (*Pan et al., 2010*). It was intriguing whether this was the same as in *Chlamydomonas*, as FLA10 and KLP20 are homologous. Thus, we tested the in vivo functionality of the chimeric motor FLA10/FLA10'/KAP. To this end, *FLA10'-HA* was transformed into an aflagellate *fla8* mutant and the transformants were expected to form a chimera with two FLA10 motor domains in vivo. FLA10/FLA10'/KAP transformants rescued the aflagellar

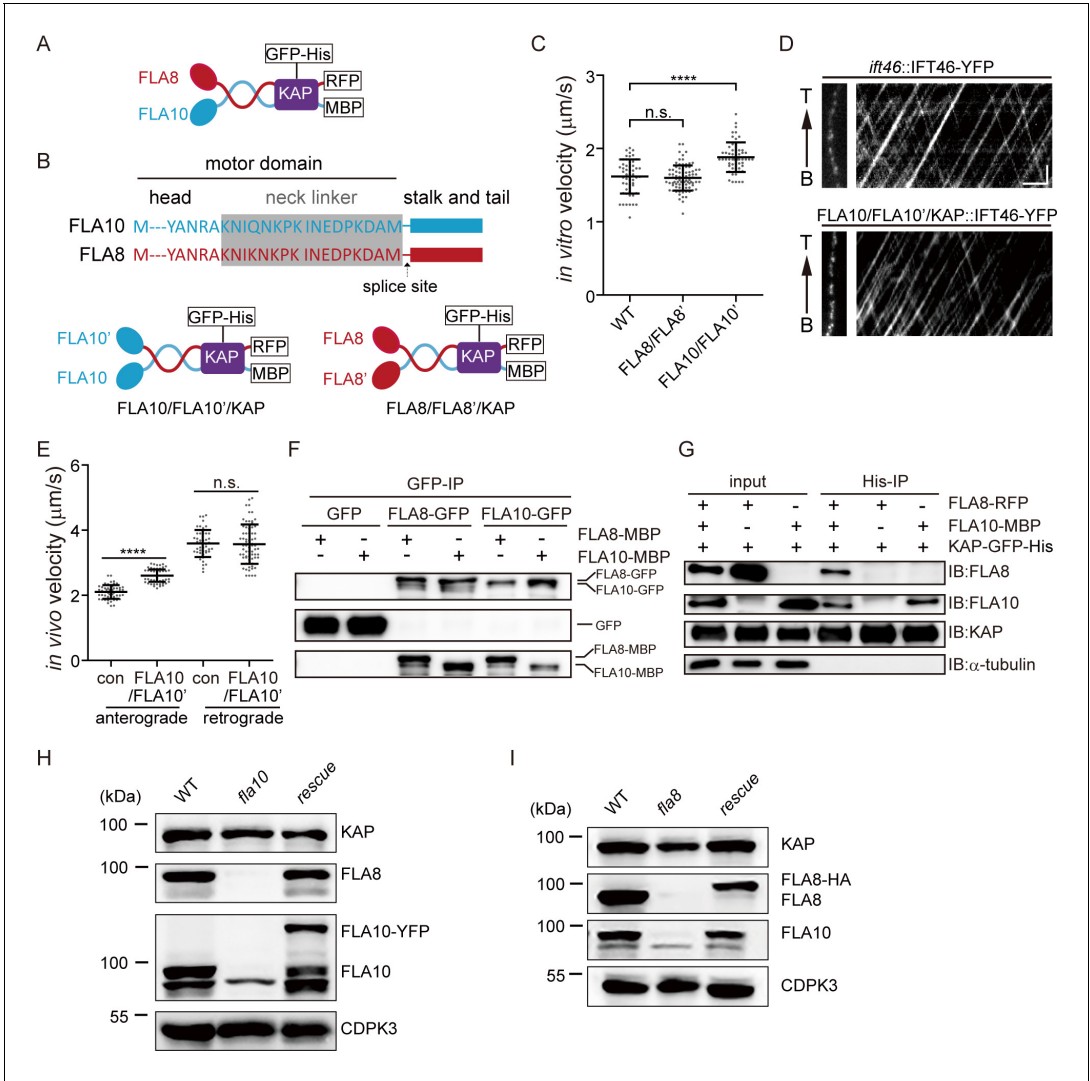

**Figure 1.** Requirement of the heterotrimeric organization of CrKinesin-II for IFT (See also *Figure 1—figure supplements 1–3*). (**A**) Schematic diagram of recombinant CrKinesin-II for expression/purification. (**B**) Overview of chimeric CrKinesin-II constructs with two identical motor domains. The motor domain of FLA10 was replaced with that of FLA8 or vice versa to create chimeric kinesin-IIs with two identical motor domains. Arrow indicates the splice site after the neck linker (gray) for creating the chimeric constructs. (**C**) In vitro motility assay of chimeric CrKinesin-IIs with two identical motor heads at 23°C. Please note, KAP is present in the chimeric motors. Data shown are mean ± SD. ****p<0.0001; n.s., statistically not significant. (**D–E**) Analysis of IFT. The velocities of IFT46-YFP expressed in FLA10/FLA10'/KAP cells or in an *ift46* rescue strain expressing IFT46-YFP (as a control) were measured using TIRF microscopy. Representative kymographs of IFT (**D**) and the measurements (**E**). Data shown are mean ± SD. ****p<0.0001; n.s., statistically not significant. (**F**) Self-interaction of FLA10 or FLA8. FLA10-GFP and FLA10-MBP or FLA8-GFP and FLA8-MBP were co-expressed respectively in 293 T cells followed by immunoprecipitation with anti-GFP antibody and immunoblotting with GFP and MBP antibodies, respectively. (**G**) FLA10 interacts with KAP while FLA8 does not. FLA10-MBP or FLA8-RFP was co-expressed respectively with KAP-GFP-His followed by pull-down with a Ni column and immunoblotting with the indicated antibodies. (**H–I**) The stability of FLA10 and FLA8 in vivo depends on each other. Cells from wild type (WT), *fla10*, and rescue (*fla10*::FLA10-YFP) (G) and cells from WT, *fla8*, rescue (*fla8*:FLA8-HA) (I) were analyzed by immunoblotting with antibodies against FLA10, FLA8, KAP and CDPK3 (as a loading control). Please note that both FLA8 and FLA10 were not detected in either *fla10* or *fla8* mutants.

The online version of this article includes the following source data and figure supplement(s) for figure 1:

**Source data 1.** Movies and numerical data for *Figure 1C*.
**Source data 2.** Movies and numerical data for *Figure 1E*.
**Figure supplement 1.** Purification of recombinant CrKinesin-II and chimeric Crkinesin-II with identical motor domains.
**Figure supplement 2.** Cells expressing chimeric CrKinesin-II with identical motor domains of FLA10 form normal cilia.
**Figure supplement 3.** Self-dimerization of FLA10 and FLA8.

phenotype of *fla8* in terms of ciliary length and ciliary regeneration kinetics (*Figure 1—figure supplement 2*), indicating that CrKinesin-II with two identical motor domains of FLA10 performs proper physiological function in vivo. To determine whether the transformants indeed rescued IFT, IFT46-YFP was expressed respectively in FLA10/FLA10'/KAP cell and an *ift46* mutant (as a control) (*Lv et al., 2017*). The retrograde IFT in FLA10/FLA10'/KAP cells (3.57 ± 0.60 μm/s, n = 70) showed a similar velocity to that in the control cells (3.59 ± 0.42 μm/s, n = 46). The velocity of anterograde IFT in FLA10/FLA10'/KAP (2.60 ± 0.19 μm/s, n = 73) was slightly higher relative to the control (2.10 ± 0.21 μm/s, n = 53) (*Figure 1D,E*). Thus, CrKinesin-II with two identical motor domains of FLA10 could function in vivo, being different from the results in *C. elegans* (*Pan et al., 2010*).

We next examined whether FLA10 or FLA8 was able to form homodimer and interact with KAP. FLA10-GFP and FLA10-MBP were co-expressed in HEK293T cells followed by immunoprecipitation with an anti-GFP antibody and immunoblotting with GFP and MBP antibodies, respectively. Similar experiments were performed for FLA8. Both FLA10 and FLA8 could self-interact (*Figure 1F*). Supposing that self-interaction of FLA10 or FLA8 can form proper homodimer, we then asked whether they could interact with KAP. FLA10-MBP and FLA8-RFP were co-expressed with KAP-GFP-His, respectively, followed by pull-down with Ni beads and immunoblotting (*Figure 1G*). Interestingly, FLA10-MBP interacted with KAP while FLA8-RFP did not. KAP is required for kinesin-II's full activation and recruitment to ciliary base (*Mueller et al., 2005*; *Sonar et al., 2020*) and FLA8 homologue KIF3B is required for the interaction of kinesin-II with IFT complex (*Funabashi et al., 2018*). Thus, neither homodimers of FLA10 or FLA8 can function in IFT, because it is likely that the FLA10 homodimer could not interact with IFT complex while FLA8 homodimer could not interact with KAP, which explains the necessity of a heterotrimeric organization of CrKinesin-II for IFT.

Earlier studies suggest that the electrostatic interactions in the neck-hinge regions prevent homodimer formation (*Chana et al., 2002*; *Rashid et al., 1995*). Thus, it is intriguing that both FLA10 and FLA8 could likely form homodimers and KIF3A can also self-dimerize (*Funabashi et al., 2018*). However, later studies with *Xenopus* kinesin-II suggest that Xklp3A can form stable homodimer while Xklp3B homodimer is less stable, and formation of heterodimer is favored when both motors are present (*De Marco et al., 2003*). Thus, self-dimerization of FLA10 and KIF3A is consistent with these data. To further validate homodimer formation of FLA10 and FLA8, we reduced the level of protein expression and found that both motors could self-dimerize (*Figure 1—figure supplement 3A,B*), suggesting that the self-dimerization is not simply due to protein over-expression. Furthermore, when FLA10 and FLA8 were both expressed, self-dimerization was suppressed though not completely (*Figure 1—figure supplement 3C and D*), which is consistent with previous study (*De Marco et al., 2003*). Next, we further tested homodimer formation in vivo. In the absence of FLA10 or FLA8 (in *fla10* and *fla8* mutants respectively) (*Liang et al., 2014*; *Matsuura et al., 2002*), the other motor could not be detected (*Figure 1H,I*). Our data suggest that even if the motor subunits can self-dimerize, they are likely to be unstable in vivo, which necessitates a functional kinesin-II with a heterodimeric motor.

## Chimeric CrKinesin-II with motor domains of HsKinesin-II functions in vitro and performs physiological function in vivo

Kinesin-II functions in various ciliated organisms to drive anterograde IFT. However, it has quite different properties. For example, the motility of kinesin-II and thus that of the anterograde IFT varies several fold between *Chlamydomonas* and mammal (*Broekhuis et al., 2014*; *Brown et al., 2015*; *Engelke et al., 2019*; *Follit et al., 2006*; *Kozminski et al., 1993*; *Muthukrishnan et al., 2009*; *Wingfield et al., 2017*). We wanted to examine whether a chimeric kinesin-II motor with motor domains from different species could perform motility functions and what would be the physiological consequences. To this end, we generated chimeric CrKinesin-IIs with one or two motor domains of human kinesin-II (*Figure 2—figure supplement 1*). Wild-type and chimeric kinesin-IIs were expressed respectively in Sf9 cells and then purified (*Figure 2A* and *Figure 2—figure supplement 1*). In vitro motility assay showed that all the chimeras could move. However, they had a similar motility to that of HsKinesin-II and was significantly slower than CrKinesin-II (~3 fold reduction) (*Figure 2B*). This result suggests that motor domains from different species can coordinate and the slower motor subunit determines the velocity of the chimeric motor.

Though the chimeric motors could function in vitro, it remains a question whether it can fulfill its physiological functions in vivo. Furthermore, the ~3 fold slower speed of the chimeric CrKinesin-II

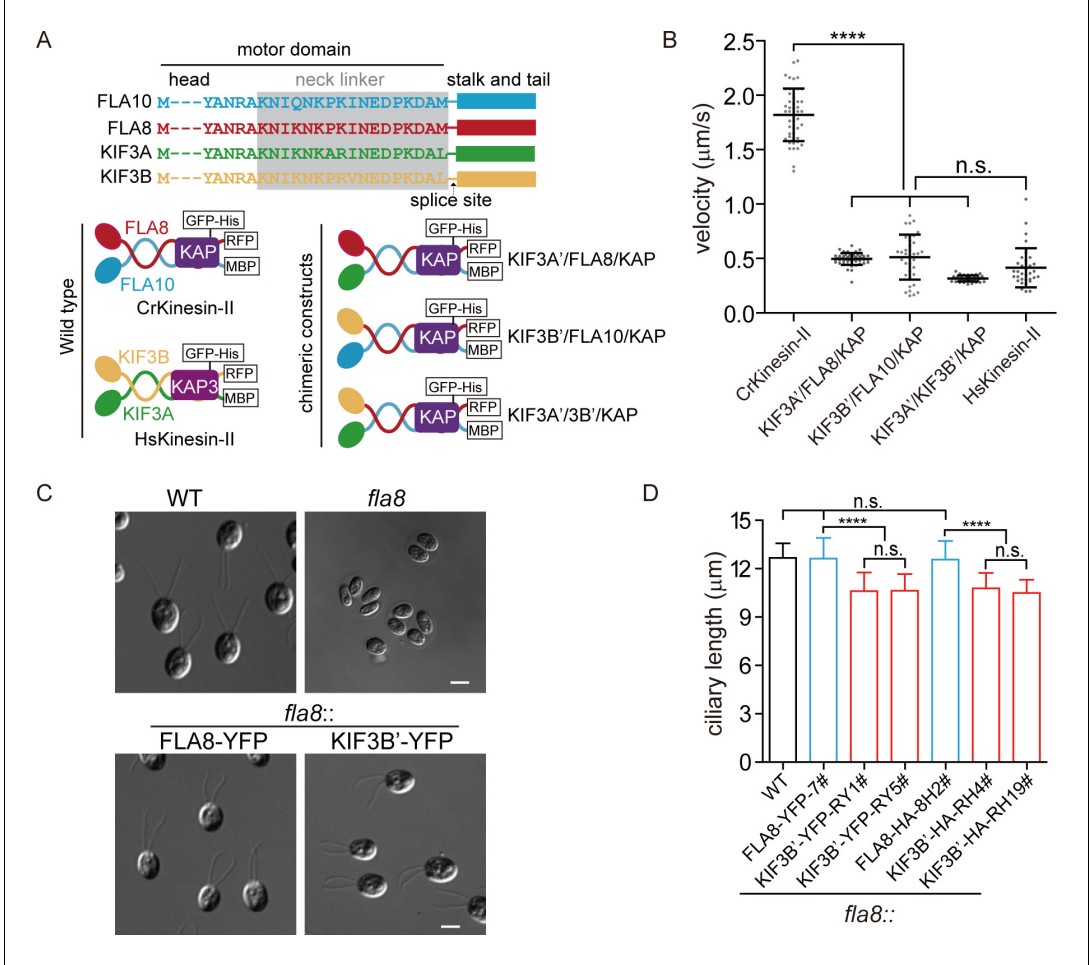

**Figure 2.** Chimeric CrKinesin-IIs with motor domains of HsKinesin-II function in vitro and in vivo (see also *Figure 2—figure supplement 1*). (A) Overview of chimeric CrKinesin-II constructs. The motor domains of FLA10, FLA8 or both in CrKinesin-II were replaced with their counterparts of HsKinesin-II, respectively. (B) In vitro motility assay of CrKinesin-II chimeras at 23°C. The rates are the following: 1.82 ± 0.24 μm/s (n = 48) for CrKinesin-II; 0.50 ± 0.05 μm/s (n = 50) for KIF3A'/FLA8/KAP; 0.51 ± 0.20 μm/s (n = 37) for KIF3B'/FLA10/KAP; 0.32 ± 0.03 μm/s (n = 40) for KIF3A'/KIF3B'/KAP and 0.41 ± 0.18 μm/s (n = 48) for HsKinesin-II. ****p<0.0001; n.s., statistically not significant. (C) Rescue of the aflagellate phenotype of *fla8* by *FLA8-YFP* or *KIF3B'-YFP*. *fla8* was transformed with *FLA8-YFP* and *KIF3B'-YFP* respectively. Cells were imaged using differential interference contrast microscopy. Wild type (WT) and *fla8* cells were shown as control. Bar, 5 μm. (D) Cells expressing KIF3B'/FLA10/KAP chimera show robust but mild decrease in ciliary length. The ciliary length in steady-state cells as indicated were measured.

The online version of this article includes the following source data and figure supplement(s) for figure 2:

**Source data 1.** Movies and numerical data for *Figure 2B*.

**Source data 2.** Representative cell images and numerical data for *Figure 2D*.

**Figure supplement 1.** Purification of recombinant HsKinesin-II and chimeric CrKinesin-II with motor domains of HsKinesin-II.

compared to the wild type (WT) CrKinesin-II would also allow us to examine how the velocity of the motor contributes to ciliary length and regeneration. We chose to test the chimeric CrKinesin-II KIF3B'/FLA10/KAP in *Chlamydomonas*. To do this, the *fla8* mutant was transformed with *KIF3B'-YFP* or *FLA8-YFP* (as a control). The transformants were expected to form KIF3B'-YFP/FLA10/KAP or FLA8-YFP/FLA10/KAP motors. Examination of the ciliary phenotype revealed that both transformants rescued the aflagellar phenotypes of *fla8* (*Figure 2C*), indicating that the chimeric KIF3B'-YFP/FLA10/KAP could function in vivo. Next, we measured ciliary length. The cilia in the KIF3B'-YFP/FLA10/KAP cells had an average length of 10.6 ± 1.1 μm (n = 50),~15% shorter compared to the control cells (12.6 ± 1.3 μm, n = 50) and WT cells (*Figure 2D*). We further verified this change by using *fla8* cells expressing *KIF3B'-HA*, which again showed ~15% reduction in length, and *FLA8-HA*, which rescues the ciliary length to the control level (*Figure 2D*). These observations demonstrate

that although the reduction in ciliary length was mild, it is a robust consequence of slow IFT mediated by a slower kinesin-II motor. Taken together, we showed that chimeric CrKinesin-II with motor domain of HsKinesin-II could function in vitro and in vivo though the chimeric motor did not fully recover the ciliary phenotype.

## Chimeric CrKinesin-II with human motor domain results in a significant reduction in IFT injection rate

The recovery of ciliary phenotype in KIF3B'-YFP/FLA10/KAP cells suggests that the chimeric motor KIF3B'-YFP/FLA10/KAP functions in anterograde IFT. Because this chimeric motor is slower in vitro than the wild-type motors, we first examined the motor velocity in vivo. Though the velocities of KIF3B'-YFP (~0.91 µm/s) and FLA8-YFP (~2.61 µm/s) were higher than their in vitro data respectively (*Figure 3A* and *Figure 2B*), KIF3B'-YFP was ~2.8 fold slower relative to FLA8-YFP, which is consistent

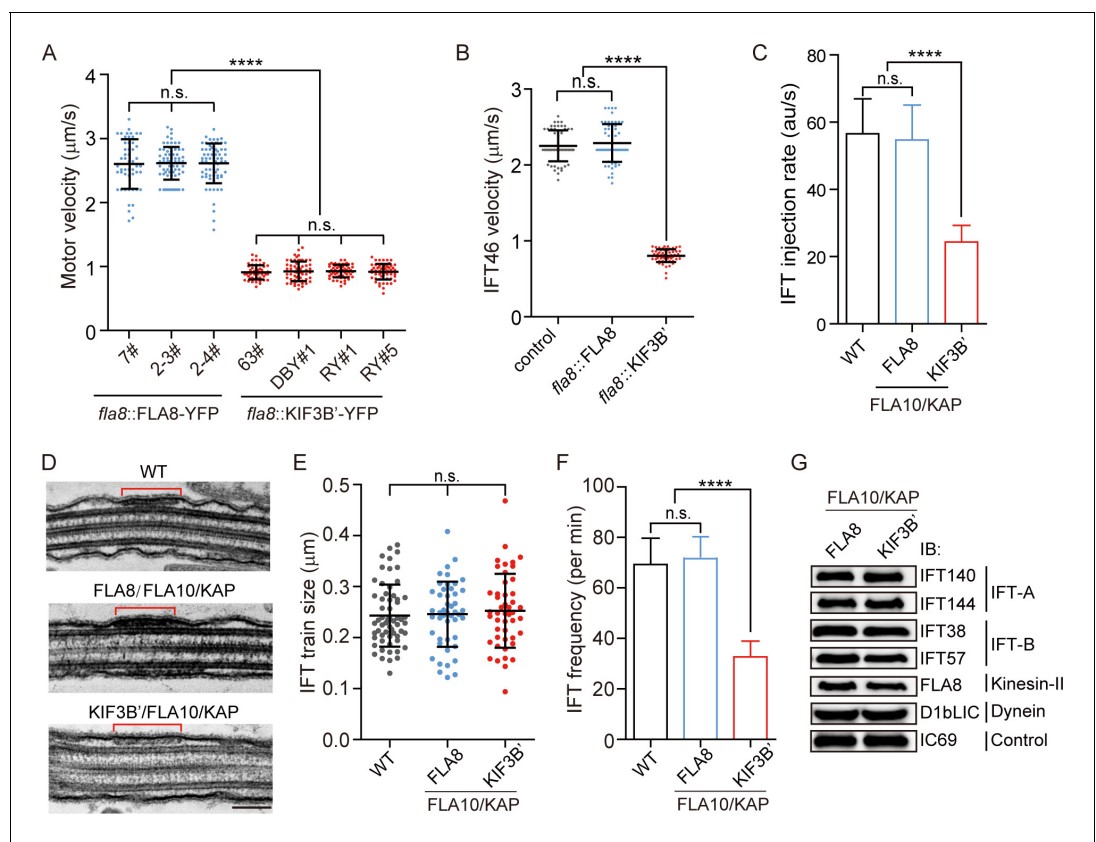

**Figure 3.** Chimeric KIF3B'/FLA10 motor leads to significant reduction in IFT injection rate but slight decrease in ciliary length. (**A**) Velocities of FLA8-YFP and KIF3B'-YFP. The anterograde velocities of FLA8-YFP and KIF3B'-YFP that were expressed respectively in *fla8* cells were assayed using TIRF microscopy. ****p<0.0001; n.s., statistically not significant. (**B**) Anterograde velocities of IFT46-YFP in FLA8 and KIF3B' transgenic cells. *IFT46-YFP* was transformed into *fla8* cells expressing FLA8-HA or KIF3B'-HA followed by analysis using TIRF microscopy. *ift46* cells expressing IFT46-YFP were used as a control. (**C**) KIF3B'/FLA10/KAP chimera leads to ~2.2-fold reduction in IFT injection rate, which was measured by monitoring fluorescence intensity of IFT46-YFP that enters into cilia per time using TIRF microscopy. (**D–E**) Analysis of the IFT train size. Representative TEM images of cilia showing IFT trains (D) and the average train size were similar among the indicated samples (E). Bar, 100 nm. (**F**) KIF3B'/FLA10 chimera leads to ~2.1-fold reduction in IFT frequency. *IFT46-YFP* was expressed in the indicated cells followed by analysis using TIRF microscopy. 69.3 ± 10.34 min⁻¹ (n = 60) for wild type (WT), 72 ± 8.2 min⁻¹ (n = 60) for FLA8-HA/FLA10/KAP and 32.7 ± 6.24 min⁻¹ (n = 60) for KIF3B'-HA/FLA10/KAP cells. (**G**) Cells expressing chimeric KIF3B'/FLA10/KAP have similar ciliary levels of IFT proteins to the controls. The cilia were isolated from the indicated cells. Equal amounts of ciliary proteins were analyzed by immunoblotting with the indicated antibodies.

The online version of this article includes the following source data for figure 3:

**Source data 1.** Movies and numerical data for *Figure 3A*.
**Source data 2.** Movies and numerical data related to *Figure 3B,C and F*.
**Source data 3.** Numerical data for *Figure 3E*.

with the data from in vitro assays (*Figure 2B*). The motor speed should reflect the velocity of antero-grade IFT. This was confirmed by measuring the velocity of an IFT protein (IFT46-YFP) in *fla8* mutants that were transformed with HA-tagged *FLA8* or *KIF3B'* (*Figure 3B*). The anterograde velocities of IFT46-YFP in the FLA8 transformant (2.29 ± 0.25 μm/s, n = 61) and KIF3B' transformant (0.80 ± 0.08 μm/s, n = 61) were similar to the velocities of the motors. Based on these results, we conclude that the chimeric CrKinesin-II KIF3B'/FLA10/KAP function in IFT but with a slower velocity.

We next analyzed how the change in motor-activity influences the ciliary entry of IFT trains into the cilium. The IFT injection rates, the amount of IFT trains entering cilia per unit time (au/s), from cells expressing chimeric and WT kinesin-II were measured. Using TIRF microscopy, we estimated IFT injection rate by monitoring the amount of IFT46-YFP entering into cilia per unit time in KIF3B'-HA/FLA10/KAP cells; FLA8-HA/FLA10/KAP and WT cells were as control. The IFT injection rate in FLA8-HA/FLA10/KAP (301.78 ± 44.21 au/s, n = 60) was similar to that in the WT cells (303.35 ± 49.42 au/s, n = 60) and was 2.2-fold of that in KIF3B'-HA/FLA10/KAP (137.57.35 ± 24.62 au/s, n = 60) (*Figure 3C*), a similar fold-change as the IFT velocity.

We then wondered how the IFT injection rate was reduced. Intuitively, IFT injection rate is the product of IFT injection frequency (number of IFT trains entering cilia per unit time) and the average size of IFT trains. Using transmission electron microscopy, we found that the average train size was similar among FLA8/FLA10/KAP, KIF3B'/FLA10/KAP, and WT cells (243–253 nm) (*Figure 3D–E*), which is consistent with a previous report (*Stepanek and Pigino, 2016*). In contrast, the IFT frequency of KIF3B'/FLA10/KAP cells was reduced by ~2.1 fold as measured by TIRF microscopy (*Figure 3F*), which is similar to the fold reduction in IFT injection rate (*Figure 3C*). Thus, we conclude that the reduction in IFT frequency accounts for the reduction in IFT injection rate in KIF3B'-HA/FLA10/KAP cells.

It is intriguing how the amount of IFT proteins inside the cilium is changed given the change in IFT injection rate and IFT velocity in the cells expressing chimeric kinesin-II. The amount of IFT protein in a cilium is given by the following equation if retrograde IFT is not considered: $M = L/v \times J$, where $M$ is the quantity of IFT proteins in a cilium; $L$ is the ciliary length (μm); $v$ is the velocity of anterograde IFT (μm/s) and $J$ is IFT injection rate ($s^{-1}$). Compared to the control cells, the ciliary length in the chimeric motor cells is about 15% shorter, the IFT injection rate and velocity were reduced ~2.8 and ~2.2 fold, respectively. Given these compensatory contributions, we predict that the ciliary levels of IFT proteins should be similar between these two cases. We performed immuno-blotting with isolated cilia and confirmed that the ciliary levels of IFT proteins were indeed similar (*Figure 3G*), supporting the above-mentioned reasoning. Taken together, we showed that the chimeric kinesin is functional in IFT though with a reduced velocity and it significantly reduces the IFT injection rate by down-regulating IFT injection frequency.

## Modeling: relationship between motor speed, ciliary assembly, and length control

The eukaryotic cilium is a model system for probing the phenomenon of organellar size control and equilibration (*Chan and Marshall, 2012*). In *Chlamydomonas*, several models have been proposed to explain ciliary length control (*Bertiaux et al., 2018b*; *Fai et al., 2019*; *Hendel et al., 2018*; *Ludington et al., 2015*; *Ma et al., 2020*; *Marshall and Rosenbaum, 2001*; *Patra et al., 2020*; *Wemmer et al., 2020*). Our experiments show that a motor with ~2.8-fold reduction in speed results in a small change in ciliary length (~15% shorter). To understand the relationship between IFT veloc-ity and ciliary length, we turned to a modeling approach. We first considered a well-established phe-nomenological model for ciliary length control (*Marshall and Rosenbaum, 2001*). In this simplest case, it is assumed that IFT limits cilia regeneration, leading to an empirical inverse scaling law (1/L) between IFT injection rate and cilium length (*Engel et al., 2009*). The reduction in IFT injection rate during ciliary elongation results in decreased ciliary assembly activity, which is eventually balanced with a constant disassembly rate and leads to a final steady-state length. The assembly rate is assumed proportional to the anterograde IFT velocity $v$ (see Materials and methods). However, this model predicts a linear and proportional dependence between the final (steady-state) ciliary length and $v$, which is inconsistent with our data (*Figure 2D*).

We reasoned that more detailed aspects of the IFT dynamics must therefore be incorporated to explain our observations, for example the role of motor diffusion in the retrograde IFT process as a possible length-sensing mechanism (*Fai et al., 2019*; *Hendel et al., 2018*; *Ma et al., 2020*).

According to the most widely accepted scenario, the cilium lengthens by addition of structural materials at the growing tip. Tubulins, the major constituent of microtubules, are transported as part of anterograde IFT cargo, which are carried by heterotrimeric kinesin-II motors. Unlike other IFT components which are recycled rapidly by active transport in both anterograde and retrograde directions, kinesin motors are returned passively by diffusion, and is a limiting resource (*Figure 4A*). The kinetics of kinesin recycling inside the *Chlamydomonas* cilium was recently confirmed experimentally via a quantitative live-cell imaging approach that followed individual IFT trains (*Chien et al., 2017*). This study revealed that the basal pool of kinesins also becomes depleted as the cilium lengthens, and the initial size of this pool has a strong effect on ciliary assembly. Here, we simulated ciliary growth based on a recent model (*Fai et al., 2019*), and compared this with our experimental data. Briefly, the model assumes a flux balance between ballistic and diffusive motor fluxes along the cilium. Over ciliary growth timescales, the total number of available motors $N = N_{ballistic} + N_{diffusive} + N_{base}$, is assumed to be conserved. Kinesin motors transport anterograde IFT trains ballistically from base to tip with speed v (µm/s), then diffuse steadily back to the base with diffusion constant D (µm²/s), where they await reinjection back into the cilium (*Figure 4A*). We consider three cases in turn, where only motors are limiting, where tubulin is also limiting, and finally the effect of a length-dependent disassembly rate (See Materials and methods).

We compared the ciliary assembly kinetics for a wild-type motor to that of a slow motor (*Figure 4B*). Parameters were chosen based on experimental data from the literature to produce a realistic growth time and a steady-state cilium length of ~12 µm when v = 2.3 µm/s (see Materials and methods). N should be the same in both WT and the mutant, as evidenced by *Figure 3G*. The ciliary regeneration kinetics predicted by the model for the two different motor speeds are consistent with our data (*Figure 4C*). The final ciliary length is reached when the assembly rate balances the disassembly rate (see Materials and methods). The fold-change in steady-state cilium length $L^{ss}/L_0^{ss}$ was evaluated over a range of different values for the fold-change in speed $v/v_0$, where $v_0$, $L_0^{ss}$ denote the wild-type motor speed and steady-state (final) cilium lengths respectively. For a reduced motor speed of $\sim v_0/3$, the model predicts a ~15% reduction in $L$ (*Figure 4D*), which agrees with our data (*Figure 2D*). Additionally the model predicts that a significantly faster motor speed would only lead to a small increase in ciliary length (saturation), if all other parameters remained unchanged.

We then explored the dependence of ciliary growth on physiological model parameters, for motors with normal versus reduced speed. We noted that in all combinations of diffusion rate and motor number that we tested, the ~3 × slower motor always leads to a mild reduction in ciliary length (*Figure 4—figure supplement 1*), thereby showing the robustness of the nonlinear and disproportional relationship between motor speed and ciliary length. We found that at the normal motor speed, the final length of cilium is diffusion-limited when motor number is constant (*Figure 4—figure supplement 1A*). In contrast, a slow motor will always lead to a cilium that is shorter than the wild-type length, regardless of the rate of diffusion unless the number of motors available in circulation is increased (i.e. motor-limited) (*Figure 4—figure supplement 1B*). Thus, a motor with slower speed would also limit IFT entry due to motor limitation. Meanwhile, there is little difference between the cilia growth timescales in the case of the wild-type motor speed, compared to the ~3 x slower motor (*Figure 4—figure supplement 1C,D*). This is again consistent with the experimental data (*Figure 4C*). In all, our results show that the slight reduction in ciliary length upon significant reduction of anterograde motor speed can be accounted for by motor limitation alone.

As an extension of our modeling analysis, we also investigated the role of tubulin-limitation (see Materials and methods: case 2), or the effect of a length-dependent disassembly rate (see Materials and methods: case 3) which was hypothesized to explain the *Chlamydomonas* length equalization phenomenon observed when one flagellum is severed (*Fai et al., 2019*). When diffusion rate (D) and number of motors (N) are varied, the corresponding phase space for the final cilium length or total growth time is qualitatively similar for physiological parameter ranges (*Figure 4—figure supplements 2* and *3*). In case 2, as the cilium lengthens and depletes the available supply of tubulin inside the cilium, the growth rate decreases – so that motor diffusion is no longer the only mechanism limiting growth (*Figure 4—figure supplement 2*). The presence of a gradient of some depolymerizer, itself undergoing a similar recycling process inside the cilium, could be the basis of a length-dependent disassembly rate (case 3) (*Figure 4—figure supplement 3*). However such a candidate depolymerizer remains to be identified in *Chlamydomonas* (*Fai et al., 2019*). While these

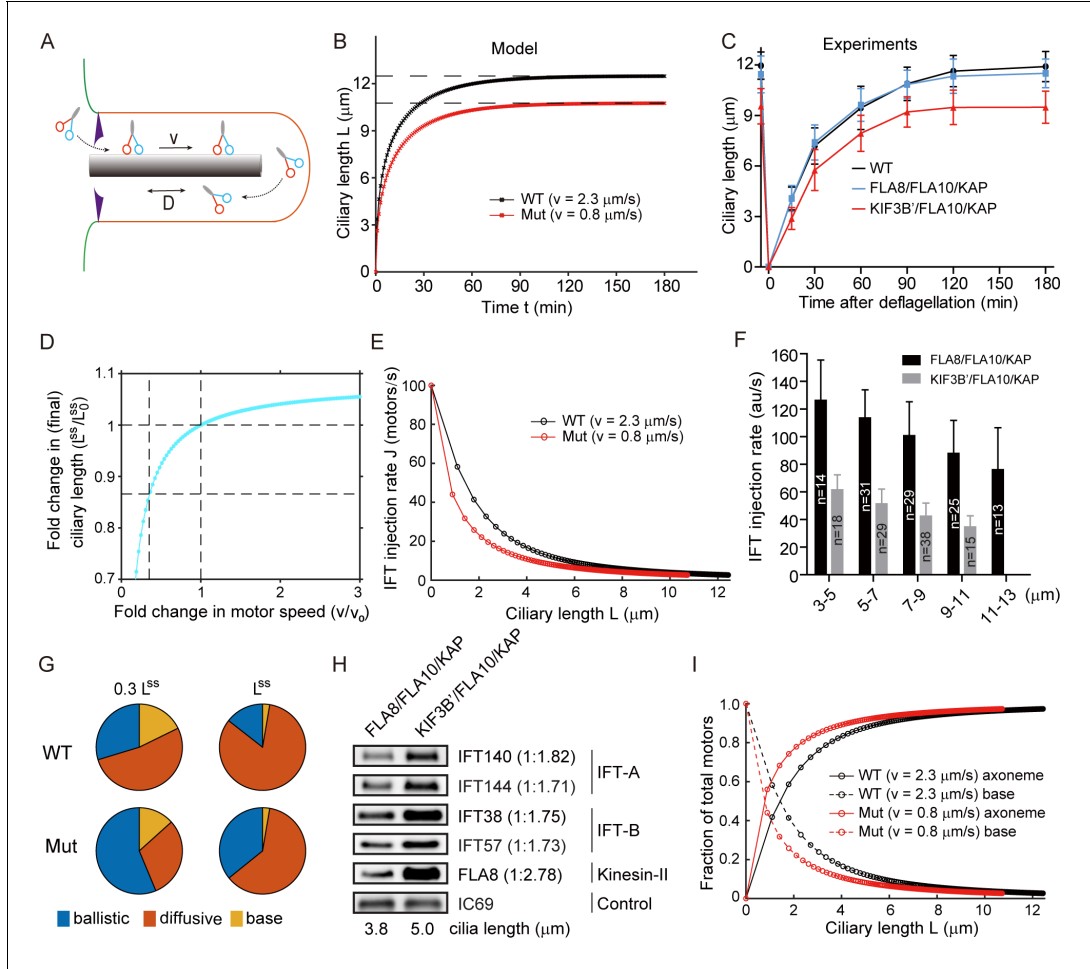

**Figure 4.** Mathematical modeling predicts a nonlinear scaling relationship between motor velocity and ciliary length (See also *Figure 4—figure supplements 1–3*). (**A**) A cartoon schematic of kinesin-II mediated IFT in cilia. (**B**) Simulated kinetics of ciliary assembly during ciliary regeneration. (**C**) Kinetics of ciliary assembly. Cells were deflagellated by pH shock to allow cilia regeneration. Cells were fixed at the indicated times followed by measurement of ciliary length. Data shown are mean ± SD (n = 50). (**D**) Modeling: relationship between motor speed and steady-state ciliary length. (**E**) Modeling: relationship between length and IFT injection rate in a growing cilium. (**F**) IFT injection rate during ciliary assembly at different length of cilia as indicated. The fluorescence intensity of IFT46-YFP was monitored via TIRF microscopy. The change in IFT injection rate from initial assembly to final length of cilia is statistically significant with p<0.0001. (**G**) Pie charts: distribution of motors at 0.3 $L_{ss}$ and $L_{ss}$. (**H**) Cells expressing slower chimeric motor KIF3B'/FLA10/KAP at shorter growing cilia exhibit higher ciliary levels of IFT proteins relative to the control. Isolated cilia during ciliary regeneration from cells as indicated were subjected to immunoblotting with the indicated antibodies. (**I**) Modeling: distribution of motors during ciliary assembly at different ciliary lengths.

The online version of this article includes the following source data and figure supplement(s) for figure 4:

**Source data 1.** Representative cells images and numerical data for *Figure 4C*, and numerical data for *Figure 4F*.

**Figure supplement 1.** Modeling (case 1): the effect of diffusion constant and the number of motors on ciliary length and the timescale to reach final ciliary length.

**Figure supplement 2.** Modeling (case 2): the effect of diffusion constant and the number of motors on ciliary length and the timescale to reach final ciliary length.

**Figure supplement 3.** Modeling (case 3): the effect of diffusion constant and the number of motors on ciliary length and the timescale to reach final ciliary length.

additional considerations can indeed modify the final cilium length and detailed regeneration kinetics, our results show that in all cases the slight reduction in ciliary length upon significant reduction of anterograde motor speed is conserved, once again demonstrating the robustness of a nonlinear dependence of ciliary length on motor speed.

Next, we simulated the IFT injection rate and motor redistribution during ciliary growth, and again compared this with our data. The model predicts that in the slow-motor mutant, the IFT

injection rate in growing cilia is lower than that in wild-type. This difference is consistent with our experimental data (*Figure 4F*). Further analysis of motor distributions predicts that when cilia are short (e.g. at 0.3 L$^{ss}$), more motors accumulate inside cilia and fewer motors are available at the base (*Figure 4G*), which, in turn, reduces the IFT injection rate (see Materials and methods *Equation 1*). We confirmed this using immunoblot and indeed, we found a higher level of IFT proteins and kinesin-2 motors inside the growing cilium of the slow-motor mutant (*Figure 4H*). Simulation results also showed that compared to wild-type, motors were redistributed in the growing cilium of the slow-motor mutant, with a higher percentage of motors in the ballistic state and less in the diffusive state (*Figure 4I*). This result supports the above-mentioned proposal that in slow-motor mutants, cilia growth became more 'motor-limited' compared to wild-type, especially in shorter growing cilia. However, we also noted that at the steady-state lengths, our simulations predict almost no difference in IFT injection rate and motor distribution between wild-type and the slow-motor mutant (*Figure 4E,G and I*). This does not agree with our experimental observations (*Figure 3C*). We think that this inconsistency is likely subject to model limitations and, in turn, would imply that more mechanistic components need to be considered in order to fully simulate cilia assembly and its underlying molecular events. Moreover, the present fluorescence microscopy techniques have limited scope for quantifying the amount of cargo loading and hence injection rate (*Wemmer et al., 2020*). These additional limitations will be discussed in the next section.

## Discussion

### Formation of a heterodimeric motor of kinesin-II and IFT

We began with the question of why kinesin-II is a motor with two heterodimeric motor subunits. Earlier studies suggested that the electrostatic interactions in the neck-hinge region of the motors prevents homodimer formation and the C-terminal stalk also has a role in heterodimer formation (*Chana et al., 2002*; *De Marco et al., 2001*; *De Marco et al., 2003*; *Rashid et al., 1995*). Studies from *De Marco et al., 2003* suggest that *Xenopus* Xklp3A can form stable homodimers in vitro while Xklp3B is less stable, heterodimerization is favored when both motors are present, which is consistent with our data where we have shown that homodimerization is suppressed in the presence of other motors. In addition, we have shown in vivo that FLA10 in the absence of FLA8 or FLA8 in the absence of FLA10 could not be detected. Thus, we propose that the formation of a heterodimeric motor for kinesin-II is likely because heterodimerization is preferred in the presence of both motor subunits, and the homodimers even if formed are likely unstable to be in vivo.

The heterodimeric motor organization of kinesin-II is also essential for its function in IFT. For this function, it requires a non-motor subunit KAP and binding of IFT complexes. KAP is required for full activation and targeting of kinesin-II to the ciliary base while KIF3B/FLA8 is required for binding the IFT complexes (*Funabashi et al., 2018*; *Mueller et al., 2005*; *Sonar et al., 2020*). Though a FLA10 homodimer was able to interact with KAP, however, a FLA10 and KAP complex could not interact with IFT complex. In contrast, a FLA8 homodimer could interact with IFT proteins, such a complex could not interact with KAP. Thus, CrKinesin-II with homodimeric motor could not function in IFT. This is also applicable to mammalian kinesin-II though due to different mechanisms. A KIF3A homodimer is able to interact with KAP but such a complex cannot interact with IFT proteins. Though KIF3B alone can interact with KAP, it cannot dimerize and thus cannot form a functional motor (*Funabashi et al., 2018*). In *C. elegans,* the two motor subunits cannot form homodimers and the particular form of the heterodimer is essential for association with KAP (*Brunnbauer et al., 2010*; *Pan et al., 2010*). Thus, the heterodimeric motor organization of kinesin-II in different organisms is essential for its function in IFT though the property of homodimer formation and the motor interaction with KAP vary. However, one should note that the reported homodimer formation was found either in vitro or by using ectopic protein expression systems. These interactions may not necessarily occur under physiological conditions, just like what we observed in *Chlamydomonas* that only heterodimer is stable in vivo. Based on this, we propose that the requirement of a heterodimeric motor for IFT is likely due to the in vivo instability of the homodimers in addition to the necessity for interaction with both the non-motor subunit KAP and the IFT complex.

## Model implication: motor limitation is a major determinant of ciliary length

Uniquely, our experimental system allowed us to evaluate how a single parameter change in motor speed affects ciliary assembly and length. Several physical models have been proposed in recent years to explain ciliary length control, but how motor speed influences ciliary length had not been experimentally evaluated. Here we adapted a recent model of ciliary length control (*Fai et al., 2019*) to understand how motor speed can affect ciliary length. These experimental and theoretical analyses show that motor speed should not significantly affect the steady-state length of cilia (*Figures 2D*, *4B and D*), and this nonlinear dependence can be accounted for by motor limitation alone. Therefore, our analyses support the previously proposed idea that motor limitation is a major determinant of ciliary growth, namely that the existence of a limited supply of motors would constrain the amount of IFT-associated proteins and cargo (e.g. tubulin) entering into cilia. This would naturally entail a decreasing ciliary assembly rate with increasing ciliary length. Finally, when ciliary assembly rate is balanced with the disassembly rate, a steady-state cilium length results (*Engel et al., 2009*; *Liang et al., 2018*; *Marshall et al., 2005*; *Marshall and Rosenbaum, 2001*; *Wemmer et al., 2020*).

Furthermore, our analyses reveal that the motor-activity-dependent anterograde journey and the diffusion-based retrograde journey make different contributions to motor limitation inside growing cilia. In steady-state, the spatial distribution of motors is equivalent to the distribution of time that the motors spend at the corresponding phase of the cyclic process. First, diffusion limits the number of motors available for IFT at steady-state lengths. The diffusion of kinesin-II from the ciliary tip to the base in *Chlamydomonas* has been demonstrated most recently by Yildiz and colleagues (with a rate of ~ 2 $\mu m^2$/s) (*Chien et al., 2017*; *Engel et al., 2009*; *Mueller et al., 2005*; *Pedersen et al., 2006*). This means that at the steady-state length (wild-type: 12 $\mu$m; the slow-motor mutant: 10.5 $\mu$m), kinesin-II motors would take ~35 s (wild-type) or ~25 s (chimeric) respectively to diffuse from ciliary tip to base. In the meantime, the anterograde journey for kinesin-II motors would be ~5 s (wild-type) or ~12 s (chimeric). These estimates suggest that at any given time in the slow-motor mutant, there would be more motors moving actively along the cilium (i.e. higher $N_{ballistic}$) because motors are slow, but fewer motors diffusing back (lower $N_{diffusive}$) because the cilium is shorter (*Figure 4I*). These two changes have opposite effects on the total number of motors inside the cilium. The final net effect is that the total time duration of kinesin-II motors inside the cilium would be similar (wild-type: 40 s; chimeric: 37 s), suggesting that the number of motors available at the base and, in turn, the IFT injection rate are similar as the cilium approaches its final length (see Materials and methods *Equation 2*). We also note that even in the slow-motor mutant, diffusion makes a greater contribution to motor limitation at steady-state lengths as the retrograde journey still takes longer than the anterograde one.

Second, the contribution of motor speed to motor limitation is most prominent in short cilia. For a wild-type cilium of 5 $\mu$m long, the anterograde and retrograde journey would both take ~2 s, each making about 50% contribution in delaying the turnover of motors. As the cilium grows longer, the time required for diffusion occupies an increasing contribution and eventually becomes the dominant factor in motor limitation at the steady-state length. In contrast, for a 5 $\mu$m cilium with the slow chimeric motors, the anterograde and retrograde journey would take ~6 s and ~2 s, respectively, showing that motor speed limits the turnover of motors to a greater extent than diffusion at short lengths. This comparison demonstrates that when motor speed is reduced, the early stage of ciliary growth is turned into a 'motor-limited' regime. This predicts that when cilia are short in the slow-motor mutant, there should be an accumulation of motors inside the cilium and a reduction in IFT injection rate. This is consistent with our experimental (*Figure 4F,H*) and simulation results (*Figure 4E,I*).

Together, our experimental and modeling supports the key role of motor limitation during ciliary growth and provide further mechanistic insights to the differential contributions of motor-based anterograde IFT and diffusion-based retrograde IFT in limiting the total number of motors available for ciliary growth. However, it should be noted that we have reduced the motor speed by using a chimeric motor, thus, we cannot absolutely exclude the possibility that this modification may lead to other potential changes (e.g. motor regulation or IFT complex assembly) that underlie the ciliary length and IFT dynamics we observed.

## Model limitations and open questions

Although the current model recapitulates the net reduction in IFT injection rate in the slow-motor mutants in growing cilia, one major limitation is that the expected level of change in our modeling results is smaller than that observed in our experiments. For example, we observe a near 2-fold reduction in IFT injection rate in growing cilia at all length ranges, while the modeling results show a decreasing fold-change in IFT injection rate and the difference is nearly diminished at the steady-state lengths. This could reflect limitations of the current model (see Materials and methods *Equation 2*). The choice of parameters dictate the IFT injection rate (particularly total number of motors [N] and the rate of diffusion [$D$]); these should not change in the slow-motor mutants, so the steady-state number of motors inside the cilium saturates to the same value in both wild-type and the slow-motor mutant. However, in our simulations, the values used for diffusion rate and motor speed are close to experimental measurements (*Chien et al., 2017*) and should reflect the true time scales of anterograde and retrograde IFT processes. Therefore, the inability of the current model to fully account for the observed change in IFT injection rate implies that more mechanistic components beyond kinesin recycling are needed to fully capture the molecular events underlying ciliary growth.

We suggest three concrete possibilities. The first is that the change of the motor domain of kinesin-II also decreases the ciliary entry of motors, thereby reducing IFT injection rate in a more direct manner. In the framework of the current model, this could be phenotypically accounted for by a reduction in the kinetic constant, K (see Materials and methods *Equation 1*). This possibility is supported by the conventional view on ciliary entry that kinesin-II carries IFT complexes with their associated cargoes into cilia (*Prevo et al., 2015*; *Rosenbaum and Witman, 2002*; *Scholey, 2013*; *Wingfield et al., 2017*). However, we also note that it was recently argued that the ciliary entry of IFT trains is mediated by a mechanism in which the motor-activity of kinesin-II is dispensable (*Nachury and Mick, 2019*). Second, negative feedback mechanisms which auto-regulate IFT injection frequency based on the amount of IFT complexes already inside the cilium are not considered in the current model (*Engel et al., 2009*; *Liang et al., 2018*). Possible candidate mechanisms may be mediated by signaling events at the ciliary base, for example those involving Ran activation or FLA8/KIF3B phosphorylation (*Liang et al., 2014*; *Ludington et al., 2013*). Theoretically, such mechanisms can be accounted for by a length-dependent K, which would further reduce the IFT injection rate, especially when more motors are accumulated inside the cilium in the short growing cilia of the slow-motor mutant (*Figure 4H*). Finally, the identity and composition of ciliary cargoes during active ciliogenesis is unclear and could be actively regulated. For instance, the limitations of fluorescence imaging for quantifying tubulin loading on IFT cargoes has been detailed previously (*Wemmer et al., 2020*). Therefore, future experimental studies are essential to further clarify and constrain these hypotheses, and thereafter motivate more detailed modeling approaches.

## Materials and methods

### Cell cultures

*Chlamydomonas* cells were cultured on 1.5% agar plates or in liquid M medium (*Sager and Granick, 1954*) at 23 °C with aeration under a 14:10 hr light-dark cycle.

### DNA constructs of chimeric kinesins for in vitro motility assay

Full-length cDNAs of KIF3A and KIFAP3 were gifts of Dr. Jiahuai Han (Xiamen University, China). Full-length cDNA of KIF3B was synthesized (WuXi Qinglan Biotech). FLA10, FLA8 and KAP cDNAs were cloned from a *Chlamydomonas* cDNA library (Takara). For chimeric kinesins, the motor domain of FLA10 was replaced with that of FLA8 or KIF3A to generate FLA8' or KIF3A' as specified in the text, respectively. The constructs for chimeric kinesins FLA10' and KIF3B' were similarly generated. The cDNAs with tags as indicated in the text were cloned in the pOCC vectors, respectively, by conventional molecular techniques.

### Protein expression and purification

Proteins used for in vitro studies were expressed in insect Sf9 cells using the baculovirus expression system. MBP-tag or His-tag at the C-terminus of the indicated proteins was used to facilitate purification while RFP-tag or GFP-tag at the C-terminus was used for imaging. The infected cells were

grown for 3 days at 27℃. Cells from 500 ml of cultures were disrupted by mortar and pestle grinding on ice in 100 ml lysis buffer (80 mM Pipes, pH 6.9; 150 mM KCl, 1 mM MgCl$_2$, 1 mM EGTA, 0.1 mM ATP, 0.1% Tween-20). The cell lysates were centrifuged at 444,000xg for 40 min and 4℃. HsKinesin-II was purified using Ni column and MBP column successively (Ni-NTA agarose affinity resin, QIA-GEN; Amylose resin, NEB New England Biolabs). For purification of Crkinesin-II, the heterodimer purified from a MBP column and KAP from a Ni column were mixed, followed by purification via Superose 6 (GE Healthcare). The proteins were frozen in liquid nitrogen and stored at −80℃.

## In vitro, single-molecule motility assay

A previously published protocol was followed for the in vitro motility assay (*Gell et al., 2010*). Briefly, 6.25 µl of 40 µM porcine brain tubulin mix containing 5% Alexa 647-labeled tubulin in BRB80 buffer with addition of 4% DMSO, 4 mM MgCl2 and 1 mM GTP (final concentrations) was incubated on ice for 5 min. Tubulins were allowed to polymerize for 2 hr at 37℃. The reaction was stopped by adding 200 µl of warm BRB80 buffer containing 20 µM taxol. Microtubules were collected in the taxol-BRB80 buffer after Airfuge centrifugation. For motility assay, the taxol stabilized microtubules were attached to a cover glass surface coated with anti-tubulin antibodies followed by the addition of indicated purified kinesins. The samples were imaged by TIRF microscopy (Olympus IX83 equipped with an Andor 897 Ultra EMCCD). The data were processed by imageJ.

## Pull-down assay

To determine possible homodimer formation of FLA10 or FLA8, cDNAs of relevant genes were cloned respectively in pEGFP-C3 vectors and were co-expressed in HEK293T cells with controls as indicated in the text. The transfected cells after growing for 48 hr were lysed in 500 µl lysis buffer (PBS, pH 7.4, 150 mM KCl, 1 mM MgCl$_2$, 1 mM EGTA, 0.1 mM ATP, 0.5% NP-40) containing protease inhibitor cocktail. After 30 min on ice, the cell lysates were centrifuged at 20, 000xg for 10 min. The supernatant was mixed with anti-GFP beads and incubated at 4℃ for 2 hr with constant rotation followed by washing with lysis buffer for three times. The samples were finally analyzed by immunoblotting with the indicated antibodies. To examine possible interactions of KAP with FLA10 or FLA8, FLA8-RFP/KAP-GFP-His or FLA10-MBP/KAP-GFP-His were co-expressed in Sf9 cells, respectively, with FLA8-RFP/FLA10-MBP/KAP-GFP-His as control. The transfected cells were lysed in lysis buffer (80 mM Pipes, pH 6.9; 150 mM KCl, 1 mM MgCl$_2$, 1 mM EGTA, 0.1 mM ATP, 0.1% Tween-20, 10 mM imidazole) containing protease inhibitor cocktail. The proteins were pulled down by Ni beads followed by washing and immunoblotting with the indicated antibodies.

## Ectopic gene expression in *Chlamydomonas*

*FLA8-HA or FLA8-YFP was* cloned in between PSAD promoter and terminator in a modified vector pKH-IFT46 (kindly provided by Dr. Kaiyao Huang, Institute of Hydrobiology) that harbors hygromycin B resistance gene. The final construct was linearized with ScaI and transformed into the *fla8* mutant by electroporation (*Liang and Pan, 2013*). The construct of KIF3B' for expression in *fla8* was made by replacing the motor domain of FLA8 with that of KIF3B. IFT46-YFP was provided by Dr. Kaiyao Huang (*Lv et al., 2017*).

## Ciliogenesis and ciliary assays

Cilia isolation or ciliary regeneration was performed as described previously (*Wang et al., 2019*; *Zhu et al., 2017b*). For ciliary regeneration, cells were deflagellated by pH shock to allow ciliary regeneration at the indicated times followed by fixation with 1% glutaraldehyde. Cells were imaged by differential interference contrast microscopy with a 40x objective on a Zeiss Axio Observer Z1 microscope (Carl Zeiss) equipped with an EM CCD camera (QuantEM512SC, Photometrics). Ciliary length from 50 cells at the indicated times was measured using ImageJ (NIH). For cilia isolation, control cells or cells during ciliary regeneration were deflagellated by pH shock. Sucrose gradient centrifugation was used to further purification of the detached cilia. Purified cilia were suspended in HMDEK buffer (50 mM HEPES, pH 7.2; 5 mM MgCl$_2$, 1 mM DTT, 0.5 mM EDTA, 25 mM KCl) containing EDTA-free protease inhibitor cocktail (mini-complete, Roche), 20 µM MG132 and 25 µg/ml ALLN, frozen in liquid nitrogen and finally stored at −80℃ until use.

## SDS-PAGE and immunoblotting

Analysis for SDS-PAGE and immunoblotting has been described previously (*Wu et al., 2018*). Cells were collected by centrifugation and lysed in buffer A (50 mM Tris-HCl, pH 7.5; 10 mM MgCl$_2$, 1 mM DTT, and 1 mM EDTA) containing EDTA-free protease inhibitor cocktail (mini-complete, Roche), 20 µM MG132 and 25 µg/ml ALLN followed by boiling in SDS sample buffer. Proteins separated on SDS-PAGE were analyzed by coomassie blue staining or immunoblotting.

Rabbit polyclonal antibodies against IFT57 and IFT38 were made by immunizing polypeptide 1–260 aa and 275-443aa, respectively, and affinity purified (Abclone, China). The other primary antibodies were detailed in the Key Resources Table. The HRP-conjugated secondary antibodies were the following: goat anti-rat, goat anti-rabbit and goat anti-mouse (1:5000, EASYBIO, China).

## Live-cell imaging of IFT

Total internal reflection fluorescence (TIRF) microscopy was used to observe live IFT. The coverslips treated with 0.01% (v/v) polylysine (Sigma) were used to immobilize cells. Images were acquired at room temperature on a Nikon microscope (A1RSi) equipped with a 100x (N.A. 1.49) TIRF objective and a cooled electron-multiplying CCD camera (Orca-flash 4.0; Hamamatsu, Japan). Images were analyzed with ImageJ (NIH, USA). The IFT speed, IFT frequency and IFT injection rate were measured following previous publications (*Engel et al., 2009*; *Wemmer et al., 2020*). The number of anterograde fluorescent IFT trains entering cilium per unit time was calculated for IFT frequency. To obtain the IFT injection rate, the fluorescence intensity of IFT trains (normalized for camera noise) per unit length of cilium and the velocity of anterograde IFT were first measured. Because most IFT trains do not stop during the transport, IFT injection rate was then calculated as the product of the fluorescence intensity and the velocity.

## Thin-section electron microscopy

Previously published protocols were followed (*Craige et al., 2010*; *Meng et al., 2014*). The samples were imaged on an electron microscope (H-7650B; Hitachi Limited) equipped with a digital camera (ATM Company).

## Modeling

We considered several existing models of ciliary length control (*Fai et al., 2019*; *Hendel et al., 2018*; *Ma et al., 2020*) with the aim of understanding the slight reduction in cilium length upon 3-fold reduction in IFT velocity. There are alternative stochastic formalisms which consider the detailed mechanism of IFT injection (*Bressloff, 2006*; *Patra et al., 2020*), but for brevity we focus only on deterministic motor kinetics here.

We adapted the model of *Fai et al., 2019* specifically for a single-flagellum. We consider three cases separately, when only motors are limiting (case 1), when both motors and tubulin are limiting (case 2), and when disassembly is also length-dependent (case 3). The latter was implicated in the *Chlamydomonas* length equalization phenomenon observed when one flagellum is severed (*Coyne and Rosenbaum, 1970*; *Fai et al., 2019*; *Rosenbaum et al., 1969*). Motors transport anterograde IFT trains with speed *v*, deposit cargoes including tubulin subunits at the growing cilium tip and then diffuse back to the base (with diffusion constant *D*). We note that the same expression for the steady-state IFT injection flux (J) appears in all three cases, a result of flux balance of kinesin motors for a limiting pool of motors (N), but the differential equation for length is slightly different.

Following *Fai et al., 2019* we assume that the IFT flux or IFT injection rate (they are the same given that IFT trains rarely stop during transport) (*J*) is proportional to the number of free motors $N_{\text{free}}$ (motors that are neither moving along the cilium nor diffusing back to the base) with a kinetic constant *K* (*Equation 1*).

$$J = K\left(N - N_{ballistic} - N_{diffusive}\right) = K\left(N - \frac{JL}{v} - \frac{JL^2}{2D}\right) \tag{1}$$

Here, the dynamics are in quasi-steady-state where the ballistic and diffusive fluxes are balanced – with the ciliary tip acting as a diffusive source and the base as a sink. This is because the timescale for transport of IFT particles over the length of the cilium (seconds) is much less than that of cilium regeneration (hours). Rearranging *Equation 1*, we obtain

$$J(v,L) = \frac{N}{\frac{1}{K} + \frac{L}{v} + \frac{L^2}{2D}} \qquad (2)$$

which reproduces the empirical finding that *J* decreases with increasing cilium length (***Engel et al., 2009***). Moreover, the formula predicts that *J* should decrease with decreasing v, when all other parameters are held constant. Both of these features are consistent with our findings (***Figure 4E,F***). Note that the scaling in ***Equation 2*** applies to any rate-limiting IFT protein (not only motors). The formalism is consistent with the measured dynamics of kinesin-II inside cilia (***Chien et al., 2017***).

## Case 1

In the simplest case, the cilium lengthens by a constant value of $\delta$ $\mu m$ for every motor. For a constant disassembly rate *d*, and J given by ***Equation 2*** above,

$$\frac{dL}{dt} = \delta J - d$$

The steady-state length is given by

$$L^{ss} = -\frac{D}{v} + \sqrt{-2D\left(\frac{1}{K} - \frac{N\delta}{d}\right) + \frac{D^2}{v^2}}$$

We simulate the system with realistic parameters N = 100, K = 1, $D = 2.5$ $\mu m^2/s$, $\delta = 1.5 \times 10^{-3}$ $\mu m$, $d = 0.004$ $\mu m/s$ to obtain 12.5 μm and 10.7 μm for wildtype and chimeric motor speeds, respectively (***Figure 4B***), and checked the dependence of steady-state length on N and D (***Figure 4—figure supplement 1***). Here, we can interpret 1/K as a delay time for motor reinjection back into the cilium.

## Case 2

If tubulin is also limiting, the assembly rate may also depend on the amount of tubulin already in circulation inside cilia. For a total tubulin pool T, we have

$$\frac{dL}{dt} = \alpha J(T - L) - d$$

where the additional constant $\alpha$ is a tubulin binding factor (constant), and *d* is a constant disassembly rate. J is the same as above. This expression is identical to ***Fai et al., 2019*** for the single-cilium case, for a constant disassembly rate. The steady-state length is given by

$$L^{ss} = -\left(\frac{D}{v} + \frac{\alpha ND}{d}\right) + \sqrt{-2D\left(\frac{1}{K} - \frac{N\alpha T}{d}\right) + \left(\frac{D}{v} + \frac{\alpha ND}{d}\right)^2}$$

With a choice of T = 30 μm (tubulin pool per flagellum), and $\alpha$ = 0.8×10$^{-4}$, we obtain similar results as before (***Figure 4—figure supplement 2***). Here, $\alpha$ has the interpretation of a tubulin binding rate constant. The phase space of N, D looks qualitatively similar (keeping other parameters fixed).

## Case 3

In ***Fai et al., 2019***, it was suggested that a length-dependent disassembly rate may be important to reproduce the biflagellar length equalization phenomenon in pairs of *Chlamydomonas* flagella (where if one flagellum is severed, the other flagellum begins to shorten as the other regrows, when lengths are equalized, both flagella increase in length once more) (***Coyne and Rosenbaum, 1970***; ***Rosenbaum et al., 1969***). Motivated by this, we further modified the length equation to examine the effect of a non-constant disassembly (with a linear concentration gradient for the depolymerizer)

$$\frac{dL}{dt} = \delta J - d - \frac{d_1 J L}{D}$$

For $d_1$ = 0.1 d, d = 0.002 μm/s we again obtain a similar dependence of motor speed on length as before. The steady-state length is given by

$$L^{ss} = -\left(\frac{D}{v} + \frac{d_1 N}{d}\right) + \sqrt{-2D\left(\frac{1}{K} - \frac{\delta N}{d}\right) + \left(\frac{D}{v} + \frac{d_1 N}{d}\right)^2}$$

The phase space of N, D is qualitatively unchanged (*Figure 4—figure supplement 3*). Variations within a realistic range of these values were found to have little effect on the overall functional dependencies, usually only resulting in faster or slower growth kinetics and/or a different final cilium length. It will be interesting to perform the single-flagellum severing experiment in the chimeric mutant to determine how length control in this context may be affected by the reduced motor speed.

## Quantification and statistical analysis

Independent experiments were carried out for at least two or more times. Data plotting was performed using Prism (GraphPad7). The data were presented as mean ± SD. Statistical significance was performed by using two-tailed Student's t test analysis. $p < 0.05$ was considered to be statistically significant. *$p < 0.05$; **$p < 0.01$; ***$p < 0.001$; ****$p < 0.0001$.

# Acknowledgements

We are grateful to Dr. Jonathon Howard (Yale University) for discussions during the course of this work. This work was supported by the National Key R and D Program of China (2018YFA0902500, 2017YFA0503500), the National Natural Science Foundation of China (31991191, 31671387, 31972888 to JP and 31922018 to XL). KYW acknowledges funding from a Springboard Award from the Academy of Medical Sciences and Global Challenges Research Fund (Genesis and control of ciliary beating, SBF003\1160).

# Additional information

### Competing interests

Junmin Pan: Reviewing editor, *eLife*. The other authors declare that no competing interests exist.

### Funding

| Funder | Grant reference number | Author |
|---|---|---|
| Ministry of Science and Technology of the People's Republic of China | 2017YFA0503500 | Junmin Pan |
| Ministry of Science and Technology of the People's Republic of China | 2018YFA0902500 | Junmin Pan |
| National Natural Science Foundation of China | 31991191 | Junmin Pan |
| National Natural Science Foundation of China | 31671387 | Junmin Pan |
| National Natural Science Foundation of China | 31972888 | Junmin Pan |
| National Natural Science Foundation of China | 31922018 | Xin Liang |
| Springboard Award from the Academy of Medical Sciences and Global Challenges Research Fund | SBF003\1160 | Kirsty Y Wan |

The funders had no role in study design, data collection and interpretation, or the decision to submit the work for publication.

## Author contributions
Shufen Li, Data curation, Formal analysis, Validation, Investigation, Visualization, Methodology, Writing - review and editing; Kirsty Y Wan, Data curation, Software, Formal analysis, Validation, Investigation, Visualization, Methodology, Writing - review and editing; Wei Chen, Hui Tao, Data curation, Formal analysis, Investigation, Methodology; Xin Liang, Conceptualization, Resources, Software, Formal analysis, Supervision, Writing - original draft, Project administration, Writing - review and editing; Junmin Pan, Conceptualization, Resources, Software, Formal analysis, Supervision, Funding acquisition, Writing - original draft, Project administration, Writing - review and editing

## Author ORCIDs
Kirsty Y Wan (iD) https://orcid.org/0000-0002-0291-328X
Wei Chen (iD) http://orcid.org/0000-0001-7454-3882
Xin Liang (iD) https://orcid.org/0000-0001-7915-8094
Junmin Pan (iD) https://orcid.org/0000-0003-1242-3791

## Decision letter and Author response
Decision letter https://doi.org/10.7554/eLife.58868.sa1
Author response https://doi.org/10.7554/eLife.58868.sa2

## Additional files

### Supplementary files
• Transparent reporting form

### Data availability
All data generated or analysed during this study are included in the manuscript and supporting files. Part of the source data have been provided for Figure 1-4.

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

# Appendix 1

**Appendix 1—key resources table**

| Reagent type (species) or resource | Designation | Source or reference | Identifiers | Additional information |
|---|---|---|---|---|
| Strain, strain background (*C. reinhardtii, mt+*) | *21gr* | Chlamdyomonas Resource Center | CC-1690 | |
| Strain, strain background (*C. reinhardtii, mt+*) | *fla8* | Chlamdyomonas Resource Center | CC-829 | |
| Strain, strain background (*C. reinhardtii, mt+*) | *fla10-2* | Chlamdyomonas Resource Center | CC-4180 | |
| Strain, strain background (*C. reinhardtii, mt+*) | FLA10/FLA10' | This study | | *fla8* transformed with a chimeric *FLA10'* with FLA10 motor domain and FLA8 tail domain |
| Strain, strain background (*C. reinhardtii, mt+*) | FLA10/FLA10'::IFT46-YFP | This study | | FLA10/FLA10' strain expressing IFT46-YFP |
| Strain, strain background (*C. reinhardtii, mt+*) | *ift46*::IFT46-YFP | This study | | *ift46* rescue strain expressing IFT46-YFP |
| Strain, strain background (*C. reinhardtii, mt+*) | wt::IFT46-YFP | This study | | Wild-type 21gr strain expressing IFT46-YFP |
| Strain, strain background (*C. reinhardtii, mt+*) | *fla8*::FLA8-YFP | This study | | *fla8* rescue strain expressing FLA8-YFP |
| Strain, strain background (*C. reinhardtii, mt+*) | *fla8*::FLA8-HA | This study | | *fla8* rescue strain expressing FLA8-HA |
| Strain, strain background (*C. reinhardtii, mt+*) | *fla8*::KIF3B'-YFP | This study | | *fla8* transformed with a YFP tagged chimeric KIF3B' having KIF3B motor domain and FLA8 tail domain |
| Strain, strain background (*C. reinhardtii, mt+*) | *fla8*::KIF3B'-HA | This study | | *fla8* transformed with an HA-tagged chimeric KIF3B' having KIF3B motor domain and FLA8 tail domain |
| Strain, strain background (*C. reinhardtii, mt+*) | *fla8*::FLA8-HA/IFT46-YFP | This study | | *fla8*::FLA8-HA strain expressing IFT46-YFP |
| Strain, strain background (*C. reinhardtii, mt+*) | *fla8*::KIF3B'-HA/IFT46-YFP | This study | | *fla8*::KIF3B'-HA strain expressing IFT46-YFP |
| Cell line (*H. sapiens*) | HEK293T | ATCC | ATCC CRL-3216 | |
| Cell line (*S. frugiperda*) | Sf9 | Expression Systems | Sf9 | |
| Transfected construct (*S. frugiperda*) | pOCC8-KAP-EGFP-His | This study | | Junmin Pan's lab |
| Transfected construct (*S. frugiperda*) | pOCC8-KAP3-EGFP-His | This study | | Same as above |

*Continued on next page*

*Appendix 1—key resources table continued*

| Reagent type (species) or resource | Designation | Source or reference | Identifiers | Additional information |
|---|---|---|---|---|
| Transfected construct (*S. frugiperda*) | pOCC25-FLA8-RFP | This study | | Same as above |
| Transfected construct (*S. frugiperda*) | pOCC25-KIF3B-RFP | This study | | Same as above |
| Transfected construct (*S. frugiperda*) | pOCC25-KIF3B'-RFP | This study | | Same as above |
| Transfected construct (*S. frugiperda*) | pOCC52-FLA10-MBP | This study | | Same as above |
| Transfected construct (*S. frugiperda*) | pOCC52-KIF3A-MBP | This study | | Same as above |
| Transfected construct (*S. frugiperda*) | pOCC52-KIF3A'-MBP | This study | | Same as above |
| Transfected construct (*H. sapiens*) | pCMV-FLA8-EGFP | This study | | Same as above |
| Transfected construct (*H. sapiens*) | pCMV-FLA10-EGFP | This study | | Same as above |
| Transfected construct (*H. sapiens*) | pCMV-FLA8-MBP | This study | | Same as above |
| Transfected construct (*H. sapiens*) | pCMV-FLA10-MBP | This study | | Same as above |
| Transfected construct (*H. sapiens*) | pCMV-FLA8 | This study | | Same as above |
| Transfected construct (*H. sapiens*) | pCMV-FLA10 | This study | | Same as above |
| Transfected construct (*H. sapiens*) | pCMV-KAP | This study | | Same as above |
| Transfected construct (*H. sapiens*) | pCMV-EGFP (pEGFP-C3) | Addgene | pEGFP-C3 | |
| Antibody | Anti-HA (rat) | Roche | clone 3F10 | 1:1000 |
| Antibody | Anti-MBP (mouse) | CMCTAG | AT0030 | 1:1000 |
| Antibody | Anti-GFP (rabbit) | Abmart | M20004S | 1:2000 |
| Antibody | Anti–α-tubulin (mouse) | Sigma-Aldrich | T6199 | 1:3000 |
| Antibody | Anti-IC69 (mouse) | Sigma-Aldrich | D6168 | 1:20000 |
| Antibody | Anti-IFT140 (rabbit) | *Zhu et al., 2017a* | | 1:2000 |
| Antibody | Anti-IFT144 (rabbit) | *Zhu et al., 2017a* | | 1:2000 |
| Antibody | Anti-IFT46 (rabbit) | *Lv et al., 2017* | | 1:2000 |
| Antibody | Anti-IFT38 (rabbit) | This study | | 1:3000 |

*Continued on next page*

*Appendix 1—key resources table continued*

| Reagent type (species) or resource | Designation | Source or reference | Identifiers | Additional information |
|---|---|---|---|---|
| Antibody | Anti-IFT57 (rabbit) | This study | | 1:2000 |
| Antibody | Anti-IFTD1bLIC (rabbit) | *Meng and Pan, 2016* | | 1:1000 |
| Antibody | Anti-FLA8 (rabbit) | *Liang et al., 2014* | | 1:3000 |
| Antibody | Anti-FLA10 (rabbit) | *Cole et al., 1998* | | 1:3000 |
| Antibody | Anti-KAP (rabbit) | *Liang et al., 2014* | | 1:3000 |
| Antibody | Anti-KIF3A (rabbit) | Abcam | ab11259 | 1:2000 |
| Antibody | Anti-KIF3B (rabbit) | Abcam | ab89278 | 1:500 |
| Antibody | Anti-KIFAP3 (KAP3) (rabbit) | Abcam | ab133537 | 1:5000 |
| Antibody | Anti-Mouse IgG (H and L)-HRP Conjugated (goat) | EasyBio | BE0102 | 1:200 |
| Antibody | Anti-Rabbit IgG (H and L)-HRP Conjugated (goat) | EasyBio | BE0101 | 1:200 |
| Antibody | Anti-Rat IgG (H+L), HRP (goat) | EasyBio | BE0108 | 1:200 |
| Antibody | Alexa Fluor 647–conjugated anti-mouse IgG (goat) | Molecular probes | A21235 | 1:200 |
| Antibody | Alexa Fluor 594–conjugated anti-rabbit IgG (goat) | Molecular probes | A11037 | 1:200 |
| Ahemical compound, drug | Ni-NTA Agarose | Qiagen | 30210 | |
| Chemical compound, drug | Amylose Resin | NEB | E8021 | |
| Chemical compound, drug | Anti-GFP Magarose Beads | Smart-life sciences | SM03801 | |
| Chemical compound, drug | Mini-Complete (EDTA-free) | Roche | 4693132001 | |
| Chemical compound, drug | MG-132 | Selleck | S2619 | |
| Chemical compound, drug | MG-101 (ALLN) | Selleck | S7386 | |
| Chemical compound, drug | Poly-lysine | Sigma-Aldrich | P8920 | |
| Software, algorithm | ImageJ | NIH Image | | https://imagej.nih.gov/ |
| Software, algorithm | GraphPad Prism 7 | GraphPad Software | | https://www.graphpad.com/ |
| Software, algorithm | Adobe Illustrator and Photoshop CS6 | Adobe | | https://www.adobe.com/Illustrator |

