## [Decision Letter]

**Acceptance summary:**

This manuscript explores the molecular determinants enabling the kinesin-II motor protein to drive intraflagellar transport (IFT) and its contribution to ciliary length control. Kinesin-II has two motor subunits and a non-motor subunit. In this study, the authors demonstrate that kinesin-II is an obligate heterodimer, as engineered homodimeric motors are either unstable or undergo protein degradation. They also showed that engineering of a slow kinesin-II mutant results in shorter flagella, which provides the missing evidence for the primary role of kinesin-II in flagellar length control in *Chlamydomonas*.

**Decision letter after peer review:**

Thank you for submitting your article "Functional Exploration of Heterotrimeric Kinesin-II in IFT and Ciliary Length Control in *Chlamydomonas*" for consideration by *eLife*. Your article has been reviewed by two peer reviewers, including Ahmet Uildiz as the Reviewing Editor and Reviewer #1, and the evaluation has been overseen by Anna Akhmanova as the Senior Editor. The reviewers have discussed the reviews with one another and the Reviewing Editor has drafted this decision to help you prepare a revised submission.

Summary:

This work explores the molecular determinants enabling kinesin-II to drive anterograde IFT and its contribution to ciliary length control. Kinesin-II homologs across species are thought to be obligate heterodimeric kinesins. In *Chlamydomonas*, the kinesin-2 complex has two heavy chains, FLA8 and FLA10, and a non-motor subunit KAP. The authors showed that both FLA8 and FLA10 tend to self-dimerize when overexpressed in cultured human cells. While FLA10 pulls down KAP, FLA8 does not. Therefore, the authors interpret this observation as FLA10 is the main interacting partner of KAP, while FLA8's main role is to connect the complex to IFT particles. As a result, neither homodimer should be biologically functional. They could not detect FLA10 protein in *fla8* deleted cells, suggesting that a homodimer is either unstable or undergoes protein degradation. The finding that a kinesin-II engineered to comprise two equal motor domains can drive IFT in *Chlamydomonas* is in principle exciting but could be better demonstrated (see below).

Perhaps, more compelling part of this manuscript begins when the authors replaced the motor domain of FLA8 with that of slower KIF3B (FLA8 homolog in humans). This chimeric motor walked slower in vitro, and rescued aflagellate phenotype in *fla8* deleted cells. Surprisingly, the slowdown of kinesin-2 driven transport led to a 15% reduction in ciliary length at equilibrium and a reduction in the IFT injection rate. Using mathematical modeling based on the balance point model, the authors try to explain the shorter flagella in the rescue strain, but the conclusions drawn need to be supported by additional experimental evidence.

Overall, the manuscript pushes the field forward by providing experimental evidence for a biophysical model developed for the flagellar length control. We can recommend the publication of this work in *eLife* if the authors sufficiently address the concerns below.

Essential revisions:

1) The presumable distinct reason for the requirement of heterodimerization in *Chlamydomonas* is not convincing: The question of why kinesin-II is a heterodimer and if this feature is required for its function in IFT in most cells and organisms is an intriguing and long-standing question in the field. The finding that an engineered motor with identical motor domains rescues a knockout strain is exciting, but the conclusions drawn need additional evidence because the claim for the distinct requirement for heterodimerization in *Chlamydomonas* is built on CoIP data. These data do not reveal if the observed interactions are biologically meaningful. Given how efficient FLA10 and FLA8 form homodimers the interaction they observe in pull-down assays, it is not clear what prevents these motors to form a significant fraction of dysfunctional homodimers in cells. Kinesin-2 may be stabilized by ternary interactions (rather than binary interactions, as slightly implied in the last paragraph of the subsection “The requirement of heterotrimeric organization of CrKinesin-II for IFT”) and self-dimerization only occurs under overexpression conditions. We recommend two ways to test this possibility. First, measure self-dimerization efficiency in the presence of KAP and the other kinesin heavy chain. Second, reduce the expression level and test if self-dimerization occurs only under overexpression conditions.

In addition, can FLA8 or FLA10 homodimers move along microtubules in vitro? What is the stoichiometry of interaction? Indeed, for vertebrate kinesin-II there are contradictory reports if KIF3A homodimers can form, but only Funabashi et al. (vs. Rashid et al., Chana et al., De Marco et al.) was cited. This section should also comprise a comprehensive discussion of why heterodimerization might be necessary for IFT function.

2) The manuscript relies too heavily on ensemble methods (pull-downs and Western blots) for quantification (such as Figure 3G), and it would greatly benefit from fluorescently tagging kinesin-2 complexes in their *Chlamydomonas* strains for quantification. It would also be highly informative how changes in kinesin-2 speed affect its flagellar localization. Rather than modeling the size of the population of diffusive and ballistically moving motors, fluorescent tagging would allow for the direct measurement of these populations. If the values agree with the model predictions, this would strengthen the interpretation that it is indeed decreased motor speed that causes the observed reduction in ciliary length.

3) The findings that kinesin-II chimeras can move, albeit at low speed, and can rescue IFT in knockout strains which possess shorter flagella are intriguing. However, the chosen mathematical modeling approach to explain the shorter cilia has several issues:

a) Modeling: The authors use the assumptions of Hendel et al., but use the model of Fai et al., who reported that the only way to reproduce all experimental findings with their model is the integration of a length-dependent change of the axoneme disassembly rate. This parameter was, however, not considered in this study.

b) Assumptions: The stated assumptions for IFT (subsection “Modeling: relationship between motor speed, ciliary assembly and length control”, last paragraph) and the limitation of the motor (or ciliary building block) as a key determinant of ciliary length (Introduction, last paragraph) are exactly what Chien et al. and Hendel et al. suggested as a key determinant.

c) Different limitation regimes: The authors postulate three motor populations: Mf (free motors at the cilium base available for binding to IFT), Mb (ballistically moving motors engaged in transporting IFT along the axoneme), and Md (motors that are diffusing back to the base of the cilium). Their modeling reveals that for the slower chimeric motors the population Mb is increased and hence the population of Mf is decreased, additionally limiting the motor. In regenerating flagella they postulate a regime switch (subsection “Motor limitation is a major determinant of ciliary length”, last paragraph) and this is based on the data in Figure 4D, E. The change in the IFT injection rate seems very small and is based on noisy data (error bars). The authors should demonstrate if this is statistically significant.

d) Non-linear scaling: The authors need to explain more clearly how the strongly reduced speed of the chimera (and injection rate) leads only to a moderately shortened flagella.

---

## [Author Response]

Essential revisions:1) The presumable distinct reason for the requirement of heterodimerization in *Chlamydomonas* is not convincing: The question of why kinesin-II is a heterodimer and if this feature is required for its function in IFT in most cells and organisms is an intriguing and long-standing question in the field. The finding that an engineered motor with identical motor domains rescues a knockout strain is exciting, but the conclusions drawn need additional evidence because the claim for the distinct requirement for heterodimerization in *Chlamydomonas* is built on CoIP data. These data do not reveal if the observed interactions are biologically meaningful. Given how efficient FLA10 and FLA8 form homodimers the interaction they observe in pull-down assays, it is not clear what prevents these motors to form a significant fraction of dysfunctional homodimers in cells. Kinesin-2 may be stabilized by ternary interactions (rather than binary interactions, as slightly implied in the last paragraph of the subsection “The requirement of heterotrimeric organization of CrKinesin-II for IFT”) and self-dimerization only occurs under overexpression conditions. We recommend two ways to test this possibility. First, measure self-dimerization efficiency in the presence of KAP and the other kinesin heavy chain. Second, reduce the expression level and test if self-dimerization occurs only under overexpression conditions.

We did these suggested experiments. The data are shown in Figure 1—figure supplement 3. Our conclusions are: (1) self-dimerization is suppressed in the presence of KAP and the other motor subunit; (2) if only one motor subunit is expressed, homodimers can still form even when the expression level is low.

Furthermore, we presented data that FLA10 was lost in a *fla8* null mutant in the last submission. In this new revision, we have presented additional data showing that FLA8 was also lost in a *fla10* null mutant. The data are shown Figure 1H, I. These data imply that even if self-dimerization could occur, the homodimers are likely not stable under physiological conditions because the stability of each motor subunit likely depends on presence of the other subunit. We think this might be the key mechanism that requires a heterodimeric motor.

In addition, can FLA8 or FLA10 homodimers move along microtubules in vitro? What is the stoichiometry of interaction? Indeed, for vertebrate kinesin-II there are contradictory reports if KIF3A homodimers can form, but only Funabashi et al. (vs. Rashid et al., Chana et al., De Marco et al.) was cited. This section should also comprise a comprehensive discussion of why heterodimerization might be necessary for IFT function.

In the current experiments, we only tested chimeric motors with only the motor domain switched, thus we do not know whether homodimer can move. Our data in vivo showed that FLA8 and FLA10 depends on each other for its stability, which reveals one mechanism why a heterodimeric motor should be formed. Because homodimers are not stable under physiological conditions, we think that it is not of great interest to further test the motility of the homodimers.

Sorry that we did miss these literatures. According to our understanding, the in vitro studies on *Xenopus* kinesin-II suggest that KIF3A and its homologues may form homodimer with a higher potential than KIF3B, and heterodimerization is favored in the presence of both motor subunits, which is consistent with our new data for competition between formation of a homodimer and a heterodimer. De Marco’s in vitro data is consistent with our data on self-dimerization of FLA10 and self-dimerization of KIF3A shown by others although self-dimerization of FLA8 is not supported. However, this may reflect some additional unknown species-specific features of the motor. For example the worm kinesin-II motors could not form homodimers (Brunnbauer et al., 2010).

All the requested literatures are cited and we have fully revised the manuscript to provide a comprehensive discussion on why heterodimeric kinesin-II is required for IFT. Please find these information in the Results subsection “The requirement of heterotrimeric organization of CrKinesin-II for IFT” and in the Discussion.

2) The manuscript relies too heavily on ensemble methods (pull-downs and Western blots) for quantification (such as Figure 3G), and it would greatly benefit from fluorescently tagging kinesin-2 complexes in their *Chlamydomonas* strains for quantification. It would also be highly informative how changes in kinesin-2 speed affect its flagellar localization. Rather than modeling the size of the population of diffusive and ballistically moving motors, fluorescent tagging would allow for the direct measurement of these populations. If the values agree with the model predictions, this would strengthen the interpretation that it is indeed decreased motor speed that causes the observed reduction in ciliary length.

We thank the reviewers for this great suggestion. Here below are our considerations and thoughts on this point.

First, using immunoblotting, we were able to quantify several IFT proteins at the same time, together with motors. In this way, our results on IFT are not only supported by quantifications on motors but also on other IFT components, which would strengthen our conclusions.

Second, we have tried to quantify the amount of motors directly using our TIRF data. However, we realized that our current imaging assay could not accurately determine the absolute amount of ballistic and diffusive motors, which may lead to inconclusive conclusions. The signal intensity of diffusive motors in retrograde journey appearing in our TIRF data was not much higher than imaging noise, which made it difficult to separate the signal of interest (diffusive motors) from the background and thereby the measurements were very noisy. Therefore, we think a better imaging method would have to be developed in our lab for presenting more accurate measurements on the amount of IFT inside cilium, which we would like to explore in future studies. For these reasons, our interpretation on the accumulation of kinesin-2 is primarily based on previous literatures (e.g. Chien et al., 2017; and Fai et al., 2019) and our own modeling analysis.

3) The findings that kinesin-II chimeras can move, albeit at low speed, and can rescue IFT in knockout strains which possess shorter flagella are intriguing. However, the chosen mathematical modeling approach to explain the shorter cilia has several issues:a) Modeling: The authors use the assumptions of Hendel et al., but use the model of Fai et al., who reported that the only way to reproduce all experimental findings with their model is the integration of a length-dependent change of the axoneme disassembly rate. This parameter was, however, not considered in this study.

We thank the reviewers for this comment. In both Hendel et al., 2018, and Fai et al., 2019, (for a single flagellum), a common assumption was that kinesin moves ballistically towards the tip and diffusively towards the base. This is also consistent with experimental results, most recently Chien et al., 2017. This is the key ingredient we wanted to adopt to explain our data.

To make this clear, we have thoroughly revised the modelling section so that three different cases are presented (see main text and Materials and methods for details). In case 3 – we considered the effect of a length-dependent change of the axonemal disassembly rate as suggested by one of the reviewers. One of the key hypotheses of the Fai et al. paper is that there is a length-dependent depolymerizer which is transported with the IFT cargo, and which also diffuses back. This was necessary to explain the two-flagella-equalization phenomenon in their model, which invoked specific assumptions about how building components, in particular motors and tubulins, are shared between the two flagella. However, length equalization could also be explained in other ways with only a constant disassembly rate (e.g. Hendel et al., 2018; Patra et al., 2020). Moreover, such a depolymerizer still has to be experimentally demonstrated in *Chlamydomonas*, as far as we know. Our new organization shows that a limited pool of motors alone is sufficient to explain the growth dynamics of a single flagellum for a chimeric motor. In future work it would be very interesting to measure length changes in the mutant when one flagellum is severed, and to further test some of the hypotheses presented in Fai et al. concerning molecular interactions between two flagella.

b) Assumptions: The stated assumptions for IFT (subsection “Modeling: relationship between motor speed, ciliary assembly and length control”, last paragraph) and the limitation of the motor (or ciliary building block) as a key determinant of ciliary length (Introduction, last paragraph) are exactly what Chien et al. and Hendel et al. suggested as a key determinant.

We thank the reviewers for making this point. We have revised the wording of the article to clarify that indeed this is consistent with the experimental evidence from Chien et al. and the model of Hendel et al. These earlier works, in combination with our own results for a speed-limited motor, serve to reinforce these concepts as the most likely mechanism for ciliary length control in *Chlamydomonas*. In the present work, our results add to this concept that both ballistic and diffusive components contribute to motor limitation, especially when cilia are short during the growing phase.

c) Different limitation regimes: The authors postulate three motor populations: Mf (free motors at the cilium base available for binding to IFT), Mb (ballistically moving motors engaged in transporting IFT along the axoneme), and Md (motors that are diffusing back to the base of the cilium). Their modeling reveals that for the slower chimeric motors the population Mb is increased and hence the population of Mf is decreased, additionally limiting the motor. In regenerating flagella they postulate a regime switch (subsection “Motor limitation is a major determinant of ciliary length”, last paragraph) and this is based on the data in Figure 4D, E. The change in the IFT injection rate seems very small and is based on noisy data (error bars). The authors should demonstrate if this is statistically significant.

We thank the reviewers for this insight – in the revised article, we provide a more extended discussion of how the decrease in injection rate arises in the chimeric motor system. In brief, we think that the distribution of motors undergoing ballistic or diffusive movement is important due to differential contributions of motor-based anterograde IFT and retrograde diffusion of motors in limiting the total number of motors available for ciliary length.

We agree with the reviewers that there are limitations with estimating IFT and/or motor number, this was a point also made in Wemmer et al., 2020. A better imaging approach is needed to more reliably quantify IFT cargo composition, which we are currently exploring in the lab. However, our data is quantitatively consistent with a previous report by Engel et al., 2009, that a gradual decrease of IFT injection rate was observed during flagellar regeneration. They have presented a scatter plot with trend overlay and demonstrated a gradual decrease (Author response image 1, blue dots for wt cells). Our data are also consistent with another publication (Liang et al., 2018). We agree that these changes are small. Because of this gradual change and limitations of fluorescence microscopy measurements, we have now calculated statistical significance between the first and last values in Figure 4F, and found they are significant, which is noted in the figure legend.

**Author response image 1. sa2fig1:** 

d) Non-linear scaling: The authors need to explain more clearly how the strongly reduced speed of the chimera (and injection rate) leads only to a moderately shortened flagella.

The non-linearity refers to the fact that the cilia are only slightly shorter despite a much slower speed. This is in contrast to the original phenomenological model by Marshall and Rosenbaum, 2001, where dL/dt = kvN/2L-d, where the final length would be linearly proportional to v. As detailed in the revised text, we think this is due to the above-mentioned regime-switch. Motor speed has the greatest effect at short flagella length, where a slower motor will take longer to complete its anterograde journey, leading to accumulation inside short flagella and reduced injection rate. However as the cilium lengthens, it very quickly becomes diffusion limited again, so that the motors spend much more in diffusive retrograde transport than it does moving actively along the tracks. This is most clearly visualized in the fold-change plot in Figure 4.